# Antimicrobial resistance patterns and molecular resistance markers of *Campylobacter jejuni* isolates from human diarrheal cases

Mohamed Elhadidy[1,2]*, Mohamed Medhat Ali[1,3], Ayman El-Shibiny[1,4], William G. Miller[5], Walid F. Elkhatib[6,7], Nadine Botteldoorn[8], Katelijne Dierick[9]

1 University of Science and Technology, Zewail City of Science and Technology, Giza, Egypt, 2 Department of Bacteriology, Mycology and Immunology, Faculty of Veterinary Medicine, Mansoura University, Mansoura, Egypt, 3 Department of Medical Microbiology and Immunology, Faculty of Medicine, Mansoura University, Mansoura, Egypt, 4 Faculty of Environmental Agricultural Sciences, Arish University, Arish, Egypt, 5 Prodce Safety and Microbiology Research Unit, Agricultural Research Service, U.S. Department of Agriculture, Albany, CA, United States of America, 6 Department of Microbiology and Immunology, School of Pharmacy & Pharmaceutical Industries, Badr University in Cairo (BUC), Entertainment Area, Badr City, Cairo, Egypt, 7 Department of Microbiology & Immunology, Faculty of Pharmacy, Ain Shams University, African Union Organization St. Abbassia, Cairo, Egypt, 8 Diergezondheidszorg Vlaanderen (DGZ), Torhout, Belgium, 9 National Reference Laboratory for *Campylobacter*, Sciensano, Scientific Service: Foodborne Pathogens, Brussels, Belgium

* melhadidy@zewailcity.edu.eg

**Data Availability Statement:** All relevant data are within the manuscript and Supporting Information files.

## Abstract

The aim of this study is to characterize the antimicrobial resistance of *Campylobacter jejuni* recovered from diarrheal patients in Belgium, focusing on the genetic diversity of resistant strains and underlying molecular mechanisms of resistance among *Campylobacter jejuni* resistant strains isolated from diarrheal patients in Belgium. Susceptibility profile of 199 clinical *C. jejuni* isolates was determined by minimum inhibitory concentrations against six commonly-used antibiotics (ciprofloxacin, nalidixic acid, tetracycline, streptomycin, gentamicin, and erythromycin). High rates of resistance were observed against nalidixic acid (56.3%), ciprofloxacin (55.8%) and tetracycline (49.7%); these rates were similar to those obtained from different national reports in broilers intended for human consumption. Alternatively, lower resistance rates to streptomycin (4.5%) and erythromycin (2%), and absolute sensitivity to gentamicin were observed. *C. jejuni* isolates resistant to tetracycline or quinolones (ciprofloxacin and/or nalidixic acid) were screened for the presence of the *tetO* gene and the C257T mutation in the quinolone resistance determining region (QRDR) of the gyrase gene *gyrA*, respectively. Interestingly, some of the isolates that displayed phenotypic resistance to these antimicrobials lacked the corresponding genetic determinants. Among erythromycin-resistant isolates, a diverse array of potential molecular resistance mechanisms was investigated, including the presence of *ermB* and mutations in the 23S rRNA gene, the *rplD* and *rplV* ribosomal genes, and the regulatory region of the *cmeABC* operon. Two of the four erythromycin-resistant isolates harboured the A2075G transition mutation in the 23S rRNA gene; one of these isolates exhibited further mutations in *rplD*, *rplV* and in the *cmeABC* regulatory region. This study expands the current understanding of how different genetic

**Funding:** BELSPO (Belgian Federal Science Policy Office) supported the research fellowship of Dr. Mohamed Elhadidy at Sciensano, Brussels.

**Competing interests:** The authors have declared that no competing interests exist.

determinants and particular clones shape the epidemiology of antimicrobial resistance in *C. jejuni* in Belgium. It also reveals many questions in need of further investigation, such as the role of other undetermined molecular mechanisms that may potentially contribute to the antimicrobial resistance of *Campylobacter*.

## Introduction

The genus *Campylobacter* currently contains 31 species, 11 subspecies and 3 biovars, several of which are relevant to human and animal health [1,2]. Among these taxa, the enteric pathogens *Campylobacter jejuni* and *Campylobacter coli* account for about 90% of human *Campylobacter* infections. An estimate of 80–90% of poultry flocks are colonized mainly with *C. jejuni*; the rest are colonized with *C. coli* or, infrequently, with *Campylobacter lari* [3–5]. Although consumption of undercooked *Campylobacter*-contaminated poultry or mishandling of raw poultry products was documented by different studies as the most important infection sources worldwide[6,7], a different recent observational studies identified other potential sources as ruminants (cattle and sheep), dogs, cats, pigs, and the environment [6,8,9].

*Campylobacter* pathogenesis is a complex process in which bacterial adherence and invasion of host cells are likely to be essential early steps [10], and a low infective dose, about 500 bacteria, is enough to induce clinical symptoms [11]. Human campylobacteriosis is considered to be one of the most crucial food-borne diseases that might challenge the future global health [1]. Over the past decade, the incidence and the prevalence of the disease exhibited a dramatic increase worldwide including North America, Europe, and Australia. Furthermore, the epidemiological data from different parts of Africa, Asia and middle east supports the epidemic profile of the disease in these areas [1].

The main symptom for *Campylobacter* infection remains acute diarrhea. However, acute inflammatory enteritis usually extends down the intestine to affect the colon and rectum [12]. In some severe cases, neurological complications such as Guillain-Barré syndrome and Miller Fisher syndrome may develop [13].

Generally, *Campylobacter* infections do not require treatment with antibiotics, as they are often of short duration, clinically mild, and self-limiting, and will usually resolve within one week after the onset of symptoms [14]. However, populations at risk (e.g., the very young, the elderly or those with underlying conditions like HIV infection) will require antibiotic treatment, which has proven to be beneficial [14]. When administered, the macrolides (such as azithromycin and erythromycin) is the drug of choice for chemotherapy in the clinical treatment of *C. jejuni* enteritis [15]. Tetracyclines and fluoroquinolones (ciprofloxacin) have been suggested as alternative drugs in the treatment of clinical campylobacteriosis, and intravenous aminoglycoside (gentamicin) injection may be required in more severe systemic cases of campylobacteriosis [16–21].

Recently, an increase in the resistance of *Campylobacter* spp. to antibiotics has been reported worldwide [22]. Contributing factors to this increase could be the veterinary usage of antibiotics as prophylaxis or for treatment of animal diseases [23]. The appearance of these resistant strains in food of animal origin is a significant public health threat [24]. In addition, recent studies have also revealed that patients infected with antimicrobial-resistant *Campylobacter* species suffer a longer duration of diarrhea, when compared with those who are infected with antimicrobial-susceptible strains [25]. Consequently, investigations of the mechanisms used by *Campylobacter* in the development of antimicrobial resistance are warranted.

In *Campylobacter*, as in most other bacterial taxa, increased antimicrobial resistance is due to spontaneous point mutations in the genome and/or the acquisition of antibiotic resistance genes or loci. In the latter scenario, most antibiotic resistance genes are plasmid-borne and acquired by *Campylobacter* via conjugation. However, the ability of *Campylobacter* spp. to acquire and incorporate DNA by natural transformation is another potential mechanism for the addition of putative antimicrobial resistance genes into the chromosome. Furthermore, some plasmids carrying drug resistance genes can integrate into the chromosome as genomic islands, thus increasing the stability of antimicrobial resistance in these strains, as reversion back to a susceptible state due to plasmid curing would be reduced. Quinolone and macrolide resistance in *Campylobacter* is usually a result of discrete and well-characterized point mutations. A single point mutation in *gyrA* (C257T) is mainly responsible for the development of resistance to quinolones and fluoroquinolones [26], while point mutations in the peptidyl encoding region of the 23S rRNA gene as well as amino acid changes in L4/L22 ribosomal proteins result in resistance to macrolides [27,28]. In *Campylobacter*, resistance to other antimicrobials, e.g. tetracycline, are due to the acquisition of resistance genes. The gene encoding the ribosomal protection protein Tet(O), which confers resistance to tetracyclines by dislodging tetracycline from its primary binding site on the ribosome [29], is generally carried on plasmids [30], although it has been identified on the chromosome [31]. Macrolide resistance in *Campylobacter* has also been shown to be due to the acquisition of the 23S rRNA methyltransferase gene *ermB* [32]. Additionally, synergistic action of the active efflux pump, CmeABC with the previously-mentioned point mutations in these genes have been reported as a well-defined resistance mechanism against different antimicrobials, including macrolides, fluoroquinolones, tetracyclines, beta-lactams, and ketolides [33].

Multi-locus sequence typing (MLST) is widely applied for studying the epidemiology of campylobacteriosis. MLST can also be used to track the dissemination of antimicrobial-resistant *C. jejuni* strains [34,35]. Different recent studies supported the clonal expansion of antimicrobial-resistant *C. jejuni* strains by revealing an association between different sequence types (STs) and antimicrobial resistance [35–40].

The aim of this study is to determine the antimicrobial resistance phenotypes in a subset of non-duplicate 199 *C. jejuni* isolates recovered from diarrheal patients in Belgium between 2006 and 2015, and to investigate the molecular mechanisms of resistance and the clonal population structure of both susceptible and resistant isolates using MLST.

## Materials and methods

### *Campylobacter* strain collection

The *C. jejuni* isolates used in in this study represent a random selection of 199 non-duplicate strains from a collection of isolates previously recovered from the stool specimens of patients suffering from different symptoms of acute gastroenteritis, including diarrhea, All isolates were obtained from the National reference center for *Campylobacter*, Saint Pierre University Hospital, Brussels, Belgium during the period between 2006 to 2015. To exclude the selection of potential clonal or epidemiological-related isolates, a stratified random-sampling scheme was performed based on year of isolation, patient location, disease severity and bacterial MLST profile.

### Media and growth conditions

Isolation of *C. jejuni* was performed as previously described [41]. Following initial isolation on Butzler selective agar (Thermo Fisher Scientific, Belgium), isolates were kept in frozen stocks at -80˚C. Frozen stocks were subcultured onto Columbia agar (Oxoid, United Kingdom)

supplemented with 5% horse blood (Sigma-Aldrich, United Kingdom) and incubated at 42˚C for 48 h under microaerobic conditions (5% $O_2$, 10% $CO_2$, 85% $N_2$) provided by the Anoxomat system (Mark II System, The Netherlands).

## Genomic DNA preparation

Extraction of genomic DNA was performed using the DNeasy Blood & Tissue Kit (Qiagen, Germany) according to the manufacturer's instructions. Eluted DNA was stored at -20˚C for further MLST screening and for molecular analysis of antimicrobial resistance (AMR) determinants.

## Antimicrobial susceptibility testing

Phenotypic screening of minimum inhibitory concentrations (MICs) to erythromycin, ciprofloxacin, nalidixic acid, gentamicin, streptomycin and tetracycline was performed as previously described using a commercial microdilution tool (Sensititre® plates; Sensititre *Campylobacter* plate–EUCAMP2, Trek Diagnostic Systems, UK) and following the manufacturer's instructions. The MIC interpretive resistance standards defined by the Clinical Laboratory Standard Institute (CLSI), 2016 were employed to define isolates resistant to ciprofloxacin, erythromycin or tetracycline. Interpretation of gentamicin, streptomycin, and nalidixic acid resistance was performed using the clinical breakpoints previously outlined [42]. *C. jejuni* strain ATCC 33560 was used as a quality control. Multidrug resistance was defined as resistance to at least three unrelated classes of antimicrobials [43].

## Molecular characterization of antimicrobial resistance

Among the resistant isolates, different potential molecular mechanisms of antibiotic resistance in *Campylobacter* were examined, including those associated with resistance to quinolones and fluoroquinolones (nalidixic acid and ciprofloxacin, respectively), macrolides (erythromycin), and tetracycline. Presence of the C257T mutation in the quinolone resistance determining region (QRDR) of *gyrA*, that potentially confers resistance to nalidixic acid and ciprofloxacin, was determined in all resistant isolates using the mismatch amplification mutation assay (MAMA-PCR) [44]. The tetracycline resistance gene *tet(O)* was detected using a previously-published PCR protocol [30]. To investigate potential mechanisms of erythromycin resistance in *C. jejuni*, isolates were screened for different genes and mutations potentially conferring resistance, including: the *ermB* gene; and mutations in the 23S rRNA gene, the *rplD* and *rplV* 50S ribosomal subunit genes, and the intergenic region between *cmeR* and *cmeABC*. PCR amplification was used to assay of the presence/absence of the *ermB* gene among resistant strains [45]. The screening of isolates for the A2074C and A2075G point mutations in the peptidyl encoding region of the 23S rRNA gene, previously reported to be highly associated with high-level erythromycin resistance, was performed using a mismatch amplification mutation protocol [46]. Sequencing of the *rplD* and *rplV* genes was performed as previously reported [17] to determine substitutions in the L4 and L22 ribosomal proteins. PCR amplification and sequencing of the intergenic region between the *cmeR* and *cmeA* genes was used to screen isolates for different polymorphisms in the regulatory region of *cmeABC*. Analysis of *cmeR* alleles and the *cme* RAIVS (*cmeR*-*cmeA* intervening sequence) region was performed using the *cmeR-ABC* locus of the erythromycin-sensitive *C. jejuni* strain NCTC 11168 as a 'wild-type' reference strain.

## DNA sequence analysis

Genes targeted for sequencing were amplified by PCR using different primer sequences and PCR amplification conditions described by the referenced authors listed above. Amplicons were purified using the QIAquick PCR Purification kit (Qiagen, Germany), following the manufacturer's instructions. Cycle sequencing reactions were performed using purified PCR products and the ABI PRISM BigDye terminator cycle sequencing kit (ver. 3.1; Life Technologies, Grand Island, NY) with standard protocols; sequencing products were purified using Big-Dye XTerminator (Life Technologies). DNA sequencing was performed on an ABI PRISM 3730 DNA Analyzer (Life Technologies), using POP-7 polymer and the ABI PRISM Genetic Analyzer Data Collection and ABI PRISM Genetic Analyzer Sequencing Analysis software. Sequences were trimmed manually and compared to those in the current databases using the BLAST suite of programs. Sequence alignments and SNP identification were performed using the Lasergene analysis package (v. 8.0.2; DNASTAR, Madison, WI).

## Clonal population structure of antimicrobial resistance

The clonal population structure of both susceptible and resistant isolates was analysed using multilocus sequence typing (MLST) as previously described [47]. Determination of allele numbers and corresponding sequence types (STs) and clonal complexes (CCs) was performed by submitting the DNA sequences to the *Campylobacter* PubMLST database website (https:// pubmlst.org/campylobacter/) at the University of Oxford.

## Data analyses

Allele sequences for each sequence type were concatenated in the order *aspA-atpA-glnA-gltA-glyA-pgm-tkt* and aligned using CLUSTALX (ver. 2.1; http://www.clustal.org/). Dendrograms were constructed using the neighbor-joining method with the Kimura 2-parameter distance estimation method [48]. Phylogenetic analyses were performed using MEGA ver. 6.06 [49] to identify evolutionary relationships between *C. jejuni* isolates. Statistical significance of the association of clonal complexes with specific antibiotic resistance was tested using a two-tailed Pearson's Chi-square test. Resistance patterns to different antibiotic classes among *C. jejuni* clonal complexes were analyzed by a two-tailed Mann–Whitney nonparametric test. Potential associations between the T86I GyrA substitution versus different clonal complexes versus ciprofloxacin resistance were evaluated by a two-tailed Pearson's Chi-square test. Correlation tests, including Kendall's tau-b, Spearman's rank correlation, and Pearson's R, were used to assess the correlations between phenotypic and genotypic patterns of clonal complexes' resistance to nalidixic acid, ciprofloxacin, and tetracycline. Data output of analyses with *p*-values less than 0.05 were considered statistically significant. Statistical analyses and descriptive statistics in the current study were performed using SPSS version 18.0 (SPSS Inc., Armonk, NY, USA).

## Ethical approval

The study represents a retrospective study that involve genotyping of historical strains collection and no patient data collection was involved in this study. Ethical approval was obtained from the respective Ethical Committee of CHU Saint Pierre.

## Results

### Antimicrobial resistance profiles

The antimicrobial resistance profiles of the assayed *C. jejuni* isolates against six different antimicrobials representing four different classes are presented in "Table 1". The highest frequency

**Table 1. Antimicrobial resistance rates of clinical *C. jejuni*.**

| Class | Antimicrobial | *C. jejuni* ATCC 33560[T] (mg/L) | MIC interpretive resistant criteria (mg/L)[a] | No. of resistant isolates (%) |
|---|---|---|---|---|
| Aminoglycosides | Gentamicin | 1 | $R \geq 8$ | 0 (0%) |
| | Streptomycin | 8 | $R \geq 4$ | 9 (4.5%) |
| Macrolides | Erythromycin | 2 | $R \geq 8$ | 4 (2%) |
| Quinolones and fluoroquinolones | Ciprofloxacin | $\leq 0.5$ | $R \geq 4$ | 111 (55.8%) |
| | Nalidixic acid | 8 | $R \geq 32$ | 112 (56.3%) |
| Tetracyclines | Tetracycline | 1 | $R \geq 16$ | 99 (49.7%) |

[a]: R: resistant

of resistance was observed for nalidixic acid (56.3%), ciprofloxacin (55.8%), and tetracycline (49.7%). A low frequency of resistance was observed for streptomycin (4.5%) and erythromycin (2%). All isolates were susceptible to gentamicin "Table 1". MIC tests yielded 12 different antimicrobial resistance patterns "Table 2". Sixty-four (32.2%) isolates were pan-susceptible to all antimicrobials tested. The most frequent antimicrobial resistance pattern observed was the combined resistance to ciprofloxacin, nalidixic acid and tetracycline (n = 71; 35.7%). Multidrug resistance was observed in 9 of 199 *C. jejuni* isolates (4.5%). Among the resistant isolates,

**Table 2. Antimicrobial resistance patterns and MLST sequence types among clinical *C. jejuni*.**

| Antimicrobial resistance profile[a] | No. of isolates n (%) | Sequence types |
|---|---|---|
| Pan susceptible | 64 (32.2%) | ST-19 (3), ST-21 (10), ST-22 (2), ST-42 (4), ST-45 (5), ST-46 (2), ST-48 (9), ST-50 (2), ST-53 (4), ST-58 (1), ST-122 (1), ST-206 (1), ST-257 (3), ST-262 (1), ST-267 (1), ST-290 (1), ST-334 (1), ST-436 (1), ST-572 (1), ST-969 (1), ST-1044 (2), ST-2187 (1), ST-2288 (1) ST-2496 (1), ST-5018, (1), ST-5222 (1), ST-5396 (1), ST-8615 (1), ST-8633 (1) |
| TET | 19 (9.5%) | ST-44 (2), ST-45 (1), ST-48 (2), ST-50 (2), ST-257 (2), ST-356 (1), ST-464 (2), ST-879 (1), ST-977 (1), ST-1707 (1), ST-5018 (1), ST-5970 (1), ST-7947 (1), ST-8634 (1) |
| CIP | 2 (1%) | ST-19 (1), ST-775 (1) |
| CIP NAL | 31 (15.6%) | ST-19 (2), ST-21 (3), ST-42 (3), ST-45 (1), ST-46 (1), ST-48 (4), ST-50 (3), ST-122 (2), ST-257 (3), ST-775 (1), ST-1044 (1), ST-1073 (1), ST-2844 (1), ST-2993 (1), ST-5018 (2), ST-7946 (1), ST-8635 (1) |
| CIP TET | 1 (0.5%) | ST-990 (1) |
| NAL STR | 1 (0.5%) | ST-572 (1) |
| STR TET | 1 (0.5%) | ST-3863 (1) |
| CIP NAL TET | 71 (35.7%) | ST-19 (1), ST-21 (3), ST-44 (5), ST-45 (2), ST-46 (2), ST-48 (7), ST-50 (4), ST-52 (1), ST-53 (1), ST-122 (2), ST-257 (1), ST-354 (2), ST-356 (1), ST-443 (1), ST-464 (8), ST-523 (2), ST-572 (4), ST-879 (1), ST-883 (2), ST-990 (2), ST-1728 (1), ST-2135 (2), ST-2254 (2), ST-2274 (6), ST-3015 (1), ST-3155 (1), ST-3546 (2), ST-3720 (2), ST-3769 (1), ST-5224 (1) |
| **<u>CIP NAL ERY TET</u>** | 2 (1%) | ST-18 (1), ST-3155 (1) |
| **<u>CIP NAL STR TET</u>** | 5 (2.5%) | ST-572 (5) |
| **<u>CIP NAL ERY STR</u>** | 1 (0.5%) | ST-5221 (1) |
| **<u>ERY NAL STR TET</u>** | 1 (0.5%) | ST-21 (1) |

[a]: MDR strains are in bold and underlined.

**Table 3. Correlation tests between phenotypic and genotypic patterns of clonal complexes' resistance to ciprofloxacin, nalidixic acid, and tetracycline.**

| Phenotypic versus genotypic resistance | Correlation test | Correlation coefficient value | Standard error | Significance* (*p*-value) |
|---|---|---|---|---|
| Ciprofloxacin resistance versus C257T *gyrA* mutation | Kendall's tau-b | 0.942 | 0.035 | $p < 0.001$ |
| | Spearman Correlation | 0.973 | 0.019 | $p < 0.001$ |
| | Pearson's R | 0.996 | 0.003 | $p < 0.001$ |
| Nalidixic acid resistance versus C257T *gyrA* mutation | Kendall's tau-b | 0.962 | 0.025 | $p < 0.001$ |
| | Spearman Correlation | 0.978 | 0.019 | $p < 0.001$ |
| | Pearson's R | 0.997 | 0.003 | $p < 0.001$ |
| Tetracycline resistance versus presence of *tet(O)* | Kendall's tau-b | 0.868 | 0.051 | $p < 0.001$ |
| | Spearman Correlation | 0.922 | 0.044 | $p < 0.001$ |
| | Pearson's R | 0.950 | 0.039 | $p < 0.001$ |

* *p*-value < 0.05 indicates a statistically significant correlation

most (n = 121; 91.0%) possessed one of three resistance profiles: TET (n = 19; 14.3%) or CIP-NAL plus (n = 71; 53.4%) or minus (n = 31; 23.3%) TET.

## Analysis of the molecular mechanisms of antibiotic resistance

The *tet(O)* gene, that potentially confers tetracycline resistance, was detected in 83 of 99 Tet[r] isolates tested (83.8%). The C257T transition in *gyrA*, which results in a T86I substitution, was observed in a total of 104 of 113 (92%) isolates that were resistant to Cip and/or Nal. Correlation tests revealed statistically-significant ($p < 0.001$), high correlation coefficients ($> 0.90$) between the C257T point mutation and resistance to ciprofloxacin as well as nalidixic acid "Table 3". Similarly, such tests showed significant ($p < 0.001$) high correlation coefficients ($> 0.85$) between tetracycline resistance and the presence of *tet(O)* "Table 3". The C257T *gyrA* mutation was absent in six isolates expressing simultaneous resistance to both ciprofloxacin and nalidixic acid and one isolate expressing resistance to nalidixic acid only. MICs in the range of 4–8 mg/L for CIP and 64 mg/L for Nal were observed in these isolates.

The four erythromycin-resistant isolates varied in their level of resistance, with three possessing an MIC of 32 mg/L and one possessing an MIC of 128 mg/L "Table 4". These isolates were screened for different resistance mechanisms including: point mutations/substitutions in the 23S rRNA, *rplD* (50S ribosomal protein L4), *rplV* (50S ribosomal protein L22) and *cmeR-ABC* loci; and the presence of *ermB*. The A2074G mutation was absent in all Erm[r] isolates; however, two of the Erm[r] isolates (CJ.12/007 and CJ.H127) harbored the A2075G mutation in the 23S rRNA gene "Table 4". In the strain demonstrating the highest level of erythromycin resistance (CJ.H127; 128 mg/L), further investigation revealed putative substitutions in the 50S

**Table 4. Minimum inhibitory concentrations (MICs), 23S rRNA gene mutations, ribosomal protein substitutions and *cmeRABC* locus alleles in four erythromycin-resistant *C. jejuni* isolates.**

| Strain | Ery MIC (mg/L) | Mutation in 23S rRNA gene | Ribosomal protein polymorphisms | | cmeR | RAIVS |
|---|---|---|---|---|---|---|
| | | | L4 (RplD) mutation | L22 (RplV) mutation | | |
| CJ.H127 | 128 | A2075G | V121A, T177S | G74A, A105M, T109A | Q118R | One bp deletion |
| CJ.11/152 | 32 | WT | WT | WT | WT | WT |
| CJ.12/007 | 32 | A2075G | WT | WT | WT | WT |
| CJ.13/164 | 32 | WT | WT | WT | WT | WT |

WT: wild type (with respect to *C. jejuni* strain NCTC 11168); RAIVS: *cmeRA* intervening sequence.

ribosomal subunit proteins L4 (V121A, T177S) and L22 (G74A, A105M, T109A). Moreover, in this strain, a putative substitution within CmeR (Q118R) and a one bp deletion within the *cme* RAIVS (*cmeR-cmeA* intervening sequence) region were identified "Table 4". No additional mutations were identified in strain CJ.12/007 and no mutations previously associated with Erm[r] were identified in strains CJ.11/152 or CJ.13/164 "Table 4". Additionally, the absence of the *ermB* gene from all erythromycin-resistant isolates was demonstrated using PCR.

## Association of MLST sequence types (STs) with antimicrobial resistance

A total of 53 different STs were represented among the 199 *C. jejuni* isolates recovered in this study "Table 2". These STs were assigned to 25 CCs and the most prevalent STs were ST-48 (n = 22; 11.0%), ST-21 (n = 17; 8.5%), ST-50 (n = 11; 5.5%), ST-572 (n = 11; 5.5%), ST-464 (n = 10; 5%), and ST-257 (n = 9; 4.5%). Twenty-six of the STs were comprised of only one isolate. All isolates from seventeen STs (ST-22, ST-58, ST-122, ST-206, ST-262, ST-267, ST-290, ST-334, ST-436, ST-969, ST-2187, ST-2288, ST-2496, ST-5222, ST-5396, ST-8615, and ST-8633) were susceptible to all antimicrobials tested "Table 2". Concerning MDR *C. jejuni* strains, only five sequence types (ST-572, ST-18, ST-21, ST-828, and ST-5221) revealed MDR patterns and more than half (5/9) of MDR prevalence was allocated into ST-572 as shown in "Table 2".

To identify the evolutionary relationships between *C. jejuni* isolates, a neighbor-joining dendrogram was constructed. For each profile identified in this study, the component allele sequences were concatenated; these concatenated sequences were aligned using CLUSTALX and a phylogenetic tree was generated using MEGA ver. 6. The relatedness of different *C. jejuni* sequence types (STs) and their associations with clonal complexes (CCs) and antimicrobial resistance patterns is depicted in Fig 1. Strains representing the three main resistance profiles (CIP-NAL, CIP-NAL-TET, and TET) are scattered throughout the dendrogram. No clustering associated with either pan-susceptibility or a particular resistance profile was observed.

## Discussion

In the few past decades, emergence of bacterial resistance to antibiotics has been a growing threat of global concern. Available antibiotics are becoming less effective and the resistance rates exceed 98% in some cases, which is not only an obstacle facing prevention and treating the disease, but also acutely increases the cost of healthcare [50]. The dramatic increase in the emergence of antimicrobial resistance observed worldwide among C. *jejuni* strains, especially to ciprofloxacin and tetracycline, has prompted investigation of the prevalence and molecular determinants of resistance. Therefore, this study aimed to provide a snapshot of the resistance phenotypes and molecular epidemiology of resistance among a collection of 199 clinical isolates obtained in Belgium over a decade (2006–2015). According to the World Health Organization, all of the antimicrobials screened in this study (with the exception of tetracycline) are considered to be critically important antimicrobials for human medicine [51].

Phenotypic screening of antimicrobial resistance frequencies among the clinical *C. jejuni* isolates tested in this study revealed high resistance rates to nalidixic acid (55.8%), ciprofloxacin (56.3%), and tetracycline (49.7%). In Belgium, similarly high resistance rates to these antimicrobials in broiler carcasses were reported in different recent epidemiological studies [35,52]. Therefore, the continued overuse of antibiotics in the case of poultry, which is the main reservoir of *Campylobacter* spp., is proposed to be responsible, at least in part, to the alarming elevated rate of AMR in Belgium. Although in the European Union, the use of

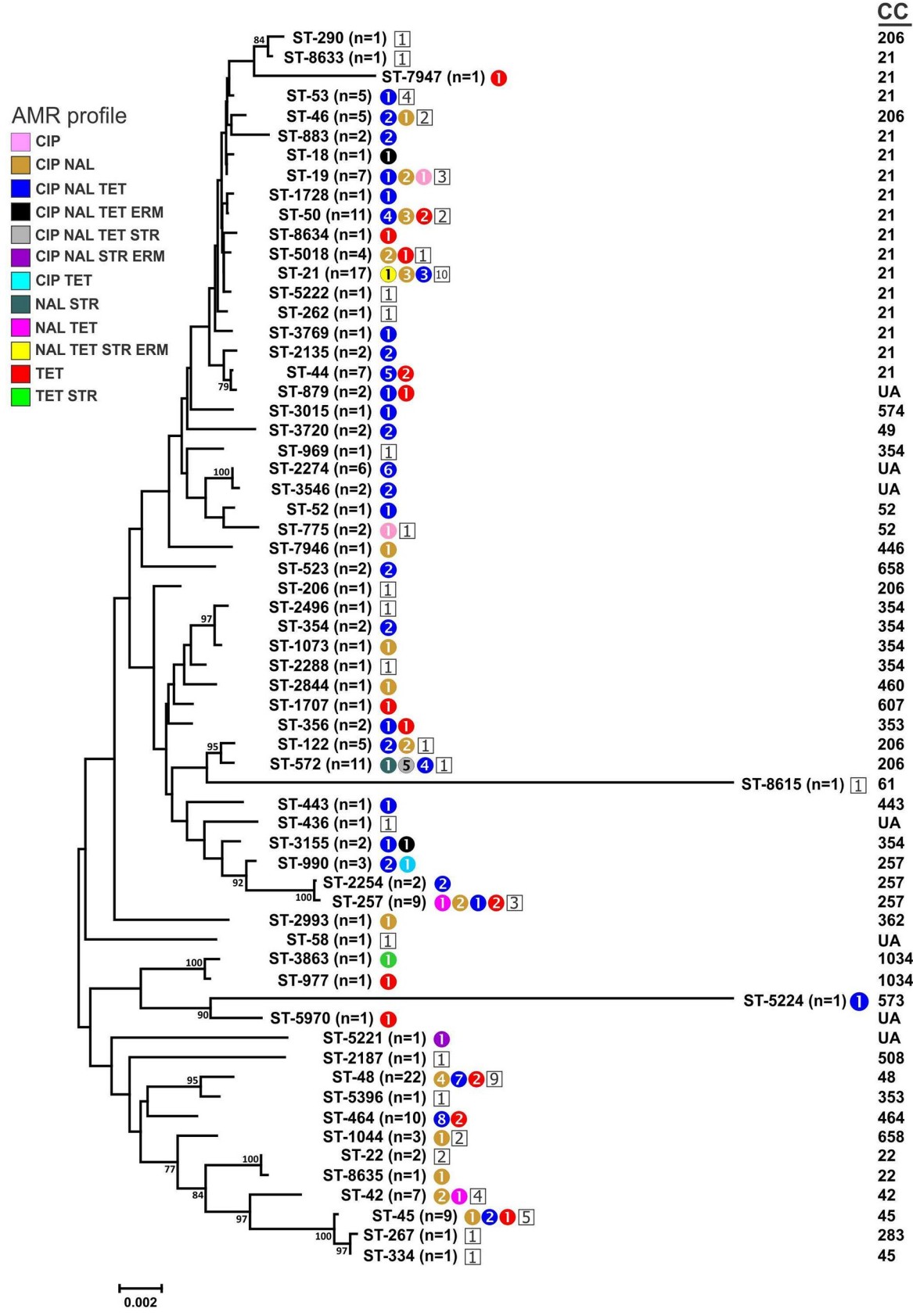

**Fig 1. Neighbor-joining dendrogram of the different *C. jejuni* sequence types (ST) and clonal complexes (CC) identified in this study.** Antimicrobial resistant strains within each ST are color-coded by profile, with the numbers of strains within each profile indicated within each circle; pan-susceptible strains within each ST are represented by a number in a white square. CIP, ciprofloxacin; ERM, erythromycin; NAL, nalidixic acid; STR, streptomycin; TET, tetracycline. UA = clonal complex unassigned.

antimicrobials as growth promotors in food-producing animals is banned and veterinary prescriptions to use antimicrobials in food animals is now required [53], some exceptions was granted to in specific cases [54]. Administration of antimicrobials via medicated feed or drinking water on a herd- or flock-wide basis and at lower doses and longer duration is a procedure usually employed during disease prophylaxis and growth promotion and is reported to increase selective pressure for antimicrobial resistance [23,55]. Moreover, the fact that poultry generally receives a higher amount of antimicrobials than other animal livestock might support the role of poultry in dissemination of antimicrobial resistance [56]. The high level of resistance observed here against tetracycline further supports the role of poultry in the dissemination of AMR to humans, since tetracycline is not commonly used in clinical medicine in Belgium, but is more routinely used for veterinary purposes as a therapeutic and preventive agent in poultry [57]. The role of poultry consumption in the antimicrobial resistance of clinical *Campylobacter* strains is further supported by different observational studies in other European countries, such as the Netherlands and Spain, and in the United States and Canada that linked the rise of fluoroquinolone-resistant *C. jejuni* infections in humans with the introduction of fluoroquinolones in poultry therapy [23,58]. A similar observation was noted in the with the emergence of fluoroquinolone-resistant *Campylobacter* following the approval of fluoroquinolones as growth promoters in veterinary practices [59]. These observational studies were corroborated by an experimental study that demonstrated the emergence of ciprofloxacin-resistant *C. jejuni* strains in broiler flocks following enrofloxacin therapy [60].

In this study, low resistance rates to streptomycin (4.5%) and erythromycin (2%) were reported, reflecting the infrequent use of these antimicrobials in clinical settings. Moreover, these results promote the use of these antimicrobials in Belgium as efficacious therapeutic agents in health care settings in lieu of other antimicrobials against which *C. jejuni* has demonstrated increased resistance, including quinolones and fluoroquinolones [61].

Multiple-drug resistance among *Campylobacter* represents an emerging trend in many developed countries that leads to drastic limitations in the selection of antimicrobial therapeutic choices [62]. In this study, 114 (57.3%) of the 199 isolates were resistant to two or more of the antimicrobials screened with nine (4.5%) exhibiting multi-drug resistance (i.e., resistance to three or more classes of antimicrobial agents). This is a higher rate than what has been reported in the European Union summary report, analyzed by European Food Safety Authority (EFSA) and European Centre for Disease Prevention and Control (ECDC) across 19 EU member states and two non-member states, where multiple drug resistance from human cases of campylobacteriosis was low (0.9%) [63]. In general, the cross-resistance to different antibiotics could arise through changes in the function of the efflux pump, such as *cmeB* that affect the susceptibility of *C. jejuni* to ampicillin, erythromycin, ciprofloxacin, and tetracycline [64] or through the acquisition of plasmids or MDR genomic island (MDRGI) [65].

In this study, the underlying molecular mechanisms of the observed resistance phenotypes were investigated. These data will provide initial and crucial information that may ultimately aid in controlling the emergence and dissemination of antimicrobial resistance. We observed a significant association between the *tet(O)* gene and tetracycline resistance and between the C257T transition in the quinolone resistance determining region (QRDR) of the *Campylobacter gyrA* gene and resistance to quinolones and fluoroquinolones. The latter mutation was absent in six *C. jejuni* isolates demonstrating simultaneous resistance to both ciprofloxacin

and nalidixic acid and one isolate exhibiting resistance to nalidixic acid only. Further research should be undertaken to explore how other modifications of the *gyrA*-encoding subunit (e.g., the Asp-203-Ser substitution), as well as mutations in the *gyrB* gene or in the efflux pumps, are potentially triggering resistance to quinolones and fluoroquinolones.

Erythromycin-resistant isolates were screened for an array of different molecular resistance mechanisms including mutations/substitutions in the 23S rRNA, *rplD*, *rplV* and *cmeRABC* loci and the presence of *ermB*. Different recent reports highlighted the crucial role of mutations at positions 2074 and 2075 in the peptidyl transferase region in domain V of the 23S rRNA target gene in the development of erythromycin resistance in *Campylobacter* [17,66,67]. Surprisingly, the A2074G mutation was not found in any of the four Erm[r] isolates. However, the A2075G mutation was found in two Erm[r] isolates, including one isolate that displayed a high minimal inhibitory concentration (MIC >128 mg/L). The presence of this mutation in a *C. jejuni* isolate with an MIC range of 32 mg/L contrasts previous reports that suggested an association between these mutations with high-level erythromycin resistance [41,46].

Further screening of amino acid changes in the L4/L22 ribosomal proteins was performed to explore additional potential mechanisms associated with erythromycin resistance among *C. jejuni*. Both proteins form portions of the polypeptide exit tunnel within the bacterial 70S ribosomal subunit, and several reports have linked substitutions in these proteins with erythromycin resistance in *C. jejuni*. Potential modifications of these proteins were only observed in one of the Erm[r] isolates (CJ.H127), that also possessed the A2075G 23S rRNA mutation. This observation might reveal an association between these modifications (perhaps also in conjunction with the A2075G mutation) and a higher level of erythromycin resistance, as the other three isolates lacking these modifications displayed lower MICs (32 mg/L). The L4 and L22 modifications identified here (V121A, T177S in the L4 protein and G74A, A105M, T109A in the L22 protein) are not located on the inside of the loop regions of these proteins (residues 55 to 77 for L4 protein and 78 to 98 for L22); however, they were previously reported to be associated with macrolide resistance in different *Campylobacter* strains [36,64] The CmeR repressor plays a crucial role in the transcriptional regulation of the efflux pump operon *cmeABC*. CmeR binds to an inverted repeat (IR) in the intervening sequence located between *cmeR* and *cmeA* [68]. It was previously elucidated that a mutation in the IR spacer reduced CmeR binding, resulting in a significant rise in phenotypic resistance to multiple antimicrobials [69]. Sequencing of the *cmeRABC* efflux pump locus identified a single amino acid substitution (Q118R) in the CmeR repressor of one erythromycin-resistant isolate (Cj.H127). Furthermore, this erythromycin-resistant isolate possessed a one bp deletion in the *cme* RAIVS (*cmeR*-*cmeA* intervening sequence) region. Further investigation on the role of different mutations in the *cme* RAIVS and their potential interactions with CmeR that lead to an overall elevation in antimicrobial resistance is needed.

The overall results of the molecular basis of macrolide resistance in *C. jejuni* highlighted the role other unreported resistance mechanisms conferring erythromycin resistance in *C. jejuni*. Future research will implement whole genome sequencing and comparative genomics studies to unravel the role of additional genetic determinants of resistance and to further analyse the interactions of CmeR at the *cmeABC* promoter region that lead to altered levels of macrolide resistance.

MLST was applied to study the genetic diversity and clonal origins of isolates tested for antimicrobial resistance. Most (80%) of the isolates assigned to ST-464 had the same resistance profile (CIP NAL TET) with all ST-464 isolates resistant to tetracycline. Association of this ST with quinolone/fluoroquinolone and tetracycline co-resistance has been previously described, supported by the observation that this ST has been reported as a fluoroquinolone-resistant lineage that has recently spread clonally in Europe [36,70]. Indeed, clonal dissemination of

AMR clones might explain, at least partially, the increasing trend of fluoroquinolone resistance observed in Europe and worldwide [37,71,72]. Fifty-three different STs were recovered from the screened isolates, thus reflecting the diversity of the AMR genotypes. Most of these sequence types including those belonging to ST 21 complex (ST 1728, ST 19, ST 19, ST 21, ST 2135, ST 262, ST 3769, ST 44, ST 50, ST 5018, 5222, ST 53, ST7974, ST883), ST 45 complex (ST 45, ST 334), ST 48 complex (ST48), ST 206 complex (ST 46, ST572, ST290, ST122, ST206) are considered as host generalists with broad host range and were previously isolated from both chickens and human clinical samples [73], supporting the hypothesis of the significant contribution of poultry to the burden of elevated antimicrobial resistance among human campylobacteriosis.

In conclusion, this study represents the molecular epidemiological investigation of resistance of clinical *C. jejuni* resistance to six different antimicrobials and the results demonstrated a current gap in the knowledge of the molecular mechanisms of resistance. Therefore, future research should attempt to both study the molecular mechanisms potentially affecting *Campylobacter* antibiotic resistance and to unravel the role of poultry meat as a potential carrier of AMR among clinical *C. jejuni* isolates. Applying a comparative genomics approach using whole-genome analysis will provide promising information regarding the molecular epidemiology of antimicrobial resistance in *C. jejuni*.

## Supporting information

**S1 Table. Epidemiological information, antimicrobial resistance and antimicrobial resistance mechanisms for reported tested strains.**
(XLSX)

## Acknowledgments

The authors are thankful to the BELSPO (Belgian Federal Science Policy Office) for supporting the research fellowship of Dr. Mohamed Elhadidy at Sciensano, Brussels. The authors are grateful to Delphine Martiny, Marie Hallin, and Olivier Vandenberg at National Reference Center for *Campylobacter*, Saint Pierre University Hospital, Brussels, Belgium for providing our group with the tested strains.

## Author Contributions

**Conceptualization:** Mohamed Elhadidy, Walid F. Elkhatib.

**Data curation:** Mohamed Elhadidy, William G. Miller.

**Formal analysis:** Mohamed Elhadidy, Mohamed Medhat Ali, Ayman El-Shibiny, William G. Miller, Walid F. Elkhatib, Nadine Botteldoorn, Katelijne Dierick.

**Investigation:** Mohamed Elhadidy.

**Methodology:** Mohamed Elhadidy.

**Writing – original draft:** Mohamed Elhadidy.

**Writing – review & editing:** Mohamed Medhat Ali, Ayman El-Shibiny, William G. Miller, Walid F. Elkhatib, Nadine Botteldoorn, Katelijne Dierick.

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
