## [Decision Letter · Decision Letter 0]

29 Nov 2019

PONE-D-19-26323

Antimicrobial Resistance Patterns and Molecular Resistance Markers of Campylobacter jejuni Isolates from Human Diarrheal Cases

PLOS ONE

Dear Dr. Elhadidy,

Thank you for submitting your manuscript to PLOS ONE. After careful consideration, we feel that it has merit but does not fully meet PLOS ONE’s publication criteria as it currently stands. Therefore, we invite you to submit a revised version of the manuscript that addresses the points raised during the review process.

The reviewer has pointed out number of areas, where clarifications and additional references are needed. These can be addressed through revision responding to each comment.

We would appreciate receiving your revised manuscript by Jan 13 2020 11:59PM. To enhance the reproducibility of your results, we recommend that if applicable you deposit your laboratory protocols in protocols.io, where a protocol can be assigned its own identifier (DOI) such that it can be cited independently in the future. For instructions see: http://journals.plos.org/plosone/s/submission-guidelines#loc-laboratory-protocols

We look forward to receiving your revised manuscript.

Kind regards,

Iddya Karunasagar

Academic Editor

PLOS ONE

Journal Requirements:

1. We ask that you please include in your methods section or as supporting information: a) The name and location of the collection from which you obtained samples for this study, such that other researchers could obtain the same strains. b) A list of the names and collection numbers of the strains used in this study.

2. Thank you for including your competing interests statement; "none"

Additional Editor Comments (if provided):

The reviewer has pointed out number of areas, where clarifications and additional references are needed. These can be addressed through revision responding to each comment.

Reviewers' comments:

Reviewer's Responses to Questions

**Comments to the Author**

1. Is the manuscript technically sound, and do the data support the conclusions?

Reviewer #1: Yes

2. Has the statistical analysis been performed appropriately and rigorously? 

Reviewer #1: Yes

3. Have the authors made all data underlying the findings in their manuscript fully available?

Reviewer #1: Yes

4. Is the manuscript presented in an intelligible fashion and written in standard English?

Reviewer #1: Yes

5. Review Comments to the Author

Reviewer #1: This research paper evaluated the prevalence of antimicrobial resistance in human Campylobacter isolates and assessed their genetic determinants. The article is well written and will provide valuable information to policy makers and researchers and will aid them to design control and prevention methods with a goal to reduce human infections with antimicrobial resistant Campylobacter.

Please find my suggestions and comments bellow.

1) Lines 55-56 – “An estimate of 80-90% of poultry flocks are colonized mainly with C. jejuni; the rest are colonized with C. coli or, infrequently, with Campylobacter lari”

I would suggest expanding this sentence and discussing in detail the source attribution of human Campylobacter infections. It is a common knowledge that consumption of poultry products is the main cause of human infections in high income countries. You should mention this fact, because it is not enough to state that poultry flocks are colonized with Campylobacter, you should link this fact with the risk they pose to human health. Moreover, you should mention other animal-origin infection sources such as ruminants (cattle or sheep). In addition, consumption of contaminated animal-origin products is not the only infection source. You should mention that human Campylobacter infections could be also caused by direct contact with food animals (i.e. cattle) or pets (dog or cat), their contaminated environment, contaminated water, or drinking raw (unpasteurized) milk.

As references you could consider these articles

“Cody AJ, Maiden MC, Strachan NJ, McCarthy ND. A systematic review of source attribution of human campylobacteriosis using multilocus sequence typing. Euro Surveill. 2019 Oct;24(43). doi: 10.2807/1560-7917.ES.2019.24.43.1800696.”

Rukambile E, Sintchenko V, Muscatello G, Kock R, Alders R. Infection, colonization and shedding of Campylobacter and Salmonella in animals and their contribution to human disease: A review. Zoonoses Public Health. 2019 Sep;66(6):562-578. doi: 10.1111/zph.12611. Epub 2019 Jun 9. Review.”

2) Line 59

As this journal has a global audience you should present the incidence of Campylobacter infections worldwide (i.e. North America, Australia etc.).

3) Line 153 – Definition of multidrug resistance requires a reference.

You should consider the following reference:

“Magiorakos AP, Srinivasan A, Carey RB, Carmeli Y, Falagas ME, Giske CG, Harbarth S, Hindler JF, Kahlmeter G, Olsson-Liljequist B, Paterson DL, RiceLB, Stelling J, Struelens MJ, Vatopoulos A, Weber JT, Monnet DL. Multidrug-resistant, extensively drug-resistant and pandrug-resistant bacteria: an international expert proposal for interim standard definitions for acquired resistance. Clin Microbiol Infect. 2012; 18:268–81.”

4. Line 127.

Please define the random sampling procedure. What was the total number of samples from which you choose 199 isolates? Why did you choose 199 isolates? Did you perform stratified random sampling considering differences in cases’ age, location (i.e. Hospital)?

5. You should mention that this study obtained the Campylobacter isolates from clinical cases, which might over estimate the prevalence of antimicrobial resistance, because these patients might have already received antimicrobial treatment.

6. Lines 302-304.

I only agree with this statement partially. I know that poultry flocks are receiving large amounts of antibacterials, however, other food animal are receiving also (i.e. swine). Moreover, you should mention that the mode of antimicrobial use has an impact on the development of resistance, and it has been shown that in-feed or in-water use of antibacterials at the whole flock / herd level has a larger impact than using individual medications. You should also mention the current antimicrobial use legislation in the EU and North America that ban the use of antibacterials as growth promoters and requires veterinary prescription for antibacterials used in food animals.

7. Line 313.

You should also consider the following reference when discussing fluoroquinolone-resistant C. jejuni infections in humans and the role of consumption of contaminated poultry products with resistant Campylobater.

“Agunos A, Léger D, Avery BP, Parmley EJ, Deckert A, Carson CA, Dutil L. Ciprofloxacin-resistant Campylobacter spp. in retail chicken, western Canada. Emerg Infect Dis. 2013 Jul;19(7):1121-4. doi: 10.3201/eid1907.111417.”

8. Line 404-408.

I think that this statement is a little bit over-reaching. This study did not present any human Campylobacter source attribution data. No information on case’s illness were provided, we do not know if these cases were consuming poultry meat or not. Moreover, poultry meat origin Campylobacter isolates were not tested in this study what makes comparison to human isolates impossible. Please revise or delete this sentence.

Thank you fro considering my suggestions.

Regards,

Csaba Varga DVM MSC PhD

Guelph, ON, Canada

6. PLOS authors have the option to publish the peer review history of their article (what does this mean?). If published, this will include your full peer review and any attached files.

Reviewer #1: Yes: Csaba Varga

---

## [Author Response · Author response to Decision Letter 0]

16 Dec 2019

The authors would like to appreciate the reviewing process and they confirm that they revised the manuscript according to all comments raised and highlighted all changes made in red.

Editorial suggestions

Can you please upload an additional copy of your revised manuscript that does not contain any tracked changes or highlighting as your main article file. 

Authors response: Revised

We ask that you please include in your methods section or as supporting information: a) The name and location of the collection from which you obtained samples for this study, such that other researchers could obtain the same strains. b) A list of the names and collection numbers of the strains used in this study.

Authors response: Revised in the material and methods and supporting information file has been added.

Reviewers suggestions: 

1) Lines 55-56 – “An estimate of 80-90% of poultry flocks are colonized mainly with C. jejuni; the rest are colonized with C. coli or, infrequently, with Campylobacter lari”

I would suggest expanding this sentence and discussing in detail the source attribution of human Campylobacter infections. It is a common knowledge that consumption of poultry products is the main cause of human infections in high income countries. You should mention this fact, because it is not enough to state that poultry flocks are colonized with Campylobacter, you should link this fact with the risk they pose to human health. Moreover, you should mention other animal-origin infection sources such as ruminants (cattle or sheep). In addition, consumption of contaminated animal-origin products is not the only infection source. You should mention that human Campylobacter infections could be also caused by direct contact with food animals (i.e. cattle) or pets (dog or cat), their contaminated environment, contaminated water, or drinking raw (unpasteurized) milk.

As references you could consider these articles

“Cody AJ, Maiden MC, Strachan NJ, McCarthy ND. A systematic review of source attribution of human campylobacteriosis using multilocus sequence typing. Euro Surveill. 2019 Oct;24(43). doi: 10.2807/1560-7917.ES.2019.24.43.1800696.”

Rukambile E, Sintchenko V, Muscatello G, Kock R, Alders R. Infection, colonization and shedding of Campylobacter and Salmonella in animals and their contribution to human disease: A review. Zoonoses Public Health. 2019 Sep;66(6):562-578. doi: 10.1111/zph.12611. Epub 2019 Jun 9. Review.”

Authors responses: The authors added revised the introduction accordingly and updates the refences as suggested.

2) Line 59

As this journal has a global audience you should present the incidence of Campylobacter infections worldwide (i.e. North America, Australia etc.).

Authors responses: The authors added revised the introduction accordingly (Lines 63-68) and updates the refences as suggested.

3) Line 153 – Definition of multidrug resistance requires a reference.

You should consider the following reference:

“Magiorakos AP, Srinivasan A, Carey RB, Carmeli Y, Falagas ME, Giske CG, Harbarth S, Hindler JF, Kahlmeter G, Olsson-Liljequist B, Paterson DL, RiceLB, Stelling J, Struelens MJ, Vatopoulos A, Weber JT, Monnet DL. Multidrug-resistant, extensively drug-resistant and pandrug-resistant bacteria: an international expert proposal for interim standard definitions for acquired resistance. Clin Microbiol Infect. 2012; 18:268–81.”

Authors responses: Suggested reference have been added (number 43)

4. Line 127.

Please define the random sampling procedure. What was the total number of samples from which you choose 199 isolates? Why did you choose 199 isolates? Did you perform stratified random sampling considering differences in cases’ age, location (i.e. Hospital)?

Authors responses: Revised accordingly

5. You should mention that this study obtained the Campylobacter isolates from clinical cases, which might over estimate the prevalence of antimicrobial resistance, because these patients might have already received antimicrobial treatment.

Authors responses: Revised accordingly

6. Lines 302-304.

I only agree with this statement partially. I know that poultry flocks are receiving large amounts of antibacterials, however, other food animal are receiving also (i.e. swine). Moreover, you should mention that the mode of antimicrobial use has an impact on the development of resistance, and it has been shown that in-feed or in-water use of antibacterials at the whole flock / herd level has a larger impact than using individual medications. You should also mention the current antimicrobial use legislation in the EU and North America that ban the use of antibacterials as growth promoters and requires veterinary prescription for antibacterials used in food animals.

Authors responses: Revised accordingly (lines 299-307)

7. Line 313.

You should also consider the following reference when discussing fluoroquinolone-resistant C. jejuni infections in humans and the role of consumption of contaminated poultry products with resistant Campylobater.

“Agunos A, Léger D, Avery BP, Parmley EJ, Deckert A, Carson CA, Dutil L. Ciprofloxacin-resistant Campylobacter spp. in retail chicken, western Canada. Emerg Infect Dis. 2013 Jul;19(7):1121-4. doi: 10.3201/eid1907.111417.”

Authors responses: Suggested reference have been added (number 58)

8. Line 404-408.

I think that this statement is a little bit over-reaching. This study did not present any human Campylobacter source attribution data. No information on case’s illness were provided, we do not know if these cases were consuming poultry meat or not. Moreover, poultry meat origin Campylobacter isolates were not tested in this study what makes comparison to human isolates impossible. Please revise or delete this sentence.

Authors responses: The sentence has been deleted

---

## [Decision Letter · Decision Letter 1]

31 Dec 2019

Antimicrobial Resistance Patterns and Molecular Resistance Markers of Campylobacter jejuni Isolates from Human Diarrheal Cases

PONE-D-19-26323R1

Dear Dr. Elhadidy,

We are pleased to inform you that your manuscript has been judged scientifically suitable for publication and will be formally accepted for publication once it complies with all outstanding technical requirements.

With kind regards,

Iddya Karunasagar

Academic Editor

PLOS ONE

Additional Editor Comments (optional):

All referee comments have been addressed satisfactorily.

Reviewers' comments:

Reviewer's Responses to Questions

**Comments to the Author**

1. If the authors have adequately addressed your comments raised in a previous round of review and you feel that this manuscript is now acceptable for publication, you may indicate that here to bypass the “Comments to the Author” section, enter your conflict of interest statement in the “Confidential to Editor” section, and submit your "Accept" recommendation.

Reviewer #1: All comments have been addressed

2. Is the manuscript technically sound, and do the data support the conclusions?

Reviewer #1: Yes

3. Has the statistical analysis been performed appropriately and rigorously? 

Reviewer #1: Yes

4. Have the authors made all data underlying the findings in their manuscript fully available?

Reviewer #1: Yes

5. Is the manuscript presented in an intelligible fashion and written in standard English?

Reviewer #1: Yes

6. Review Comments to the Author

Reviewer #1: (No Response)

7. PLOS authors have the option to publish the peer review history of their article (what does this mean?). If published, this will include your full peer review and any attached files.

Reviewer #1: Yes: Csaba Varga

---

## [Editor Report · Acceptance letter]

9 Jan 2020

PONE-D-19-26323R1 

Antimicrobial Resistance Patterns and Molecular Resistance Markers of *Campylobacter jejuni* Isolates from Human Diarrheal Cases

Dear Dr. Elhadidy:

I am pleased to inform you that your manuscript has been deemed suitable for publication in PLOS ONE. Congratulations! Your manuscript is now with our production department. 

With kind regards,

on behalf of

Dr. Iddya Karunasagar 

Academic Editor

PLOS ONE